# Perceived water-related risk factors of Buruli ulcer in two villages of south-central Côte d'Ivoire

**Andrea Leuenberger** [1,2] *, **Bognan V. Koné**[3], **Raymond T. A. S. N'krumah**[3,4], **Didier Y. Koffi**[3,5], **Bassirou Bonfoh**[3], **Jürg Utzinger**[1,2], **Gerd Pluschke**[1,2]

**1** Swiss Tropical and Public Health Institute, Allschwil, Switzerland, **2** University of Basel, Basel, Switzerland, **3** Centre Suisse de Recherches Scientifiques en Côte d'Ivoire, Abidjan, Côte d'Ivoire, **4** Université Peleforo Gon Coulibaly de Korhogo, Korhogo, Côte d'Ivoire, **5** Programme National de Lutte contre l'Ulcère de Buruli, Abidjan, Côte d'Ivoire

* andrea.leuenberger@swisstph.ch

**Data Availability Statement:** Data cannot be shared publicly because of third party restrictions. Requests to access the data that support the findings of the study can be submitted on https://

## Abstract

### Background

Buruli ulcer, caused by *Mycobacterium ulcerans*, is a neglected tropical skin disease that is primarily endemic in West and Central Africa, including Côte d'Ivoire. Studies indicate that *M. ulcerans* infections are caused by contact with an environmental reservoir of the bacteria, governed by specific human biological conditions. Yet, the nature of this reservoir and the exact mode of transmission remain unknown.

### Methodology

To identify ecologic risk factors of Buruli ulcer in south-central Côte d'Ivoire, we pursued a qualitative study matched with geo-referencing inquiry. Embedded in a broader integrated wound management research project, we (i) mapped households and water sources of laboratory confirmed Buruli ulcer cases and (ii) interviewed 12 patients and four health care workers to assess exposure to surface water and to deepen the understanding of perceived transmission pathways.

### Principal findings

Water availability, accessibility, and affordability were reported as key determinants for choosing water resources. Furthermore, perceived risks were related to environmental, structural, and individual factors. Despite the presence of improved water sources (e.g., drilled wells), communities heavily relied on unprotected surface water for a multitude of activities. The nearby Bandama River and seasonal waterbodies were frequently used for washing, bathing, and collection of water for drinking and cooking. Many residents also reported to cross the river on a daily basis for agricultural chores, and hence, are exposed to stagnant water during farming activities.

doi.org/10.5281/zenodo.7314078. In agreement with project partners, data will be made available.

**Funding:** The author(s) received no specific funding for this work.

**Competing interests:** The authors have declared that no competing interests exist.

## Conclusions/significance

Our study in two Buruli ulcer endemic villages in south-central Côte d'Ivoire revealed a wide range of water-related domestic activities that might expose people to an increased risk of contracting the disease. Environmental, biological, social, and cultural risk factors are closely interlinked and should be considered in future investigations of Buruli ulcer transmission. Active participation of the communities is key to better understand their circumstances to advance research and fight against Buruli ulcer and other neglected tropical diseases.

## Author summary

Buruli ulcer is a bacterial skin disease, which is endemic in West and Central Africa. The disease mostly affects children and can progress to large lesions primarily located on upper and lower limbs. While current evidence suggests that the causative bacterium, *Mycobacterium ulcerans*, is hosted in aquatic environments, the exact transmission route is still not clear. In our study, we investigated two endemic villages in the south-central part of Côte d'Ivoire. Framed in a larger research initiative, we mapped households and water sources of Buruli ulcer cases. Additionally, we interviewed people with Buruli ulcer as well as health care workers about water contact patterns and perceived risk factors. Despite drilled wells installed in the villages, the nearby river (Bandama), seasonal ponds, and low-lands are key water resources for the residents. Our research revealed a variety of activities where people get in touch with water that are related to domestic and subsistence work. Environmental, social, and cultural risk factors were mentioned, underlining the complexity on the local level. We conclude that the communities' perspective and their knowledge must be considered in future research to reduce the burden of Buruli ulcer and other neglected tropical diseases.

## Introduction

Buruli ulcer is a neglected tropical skin disease caused by the bacterium *Mycobacterium ulcerans* [1], which is primarily endemic in West and Central Africa [2–4]. Populations with no or only poor access to clean water and improved sanitation are at highest risk with children making up more than 50% of all cases [3,5,6]. To reduce the burden of Buruli ulcer, integrated approaches for disease management have been suggested, emphasizing diagnosis and treatment of wounds early [7,8]. Despite this recommendation, the situation in endemic settings is complex, partially explained by limited financial and human resources in the public health systems and water infrastructure [9]. In remote rural areas, patients often arrive with large lesions, which can amplify life-long secondary implications and require skilled medical staff for management [10,11]. Traditional healers have often remained the preferred focal point and first line for treatment, partially related to the conception that Buruli ulcer has a supernatural etiology [12,13]. Additionally, chronic wounds are culturally stigmatizing and general knowledge about Buruli ulcer and other skin diseases in affected populations is poor, which calls for tailored information, education, and communication (IEC) [14]. However, disease-specific IEC remains challenging as the exact mode of transmission of *M. ulcerans* is still unknown [15–18].

*M. ulcerans* is considered an environmental pathogen, while human-to-human spread is thought to play no significant role. Of note, *M. ulcerans* has a doubling time of more than 24

hours making culture difficult. To date only two reports on the successful primary isolation of *M. ulcerans* strains from the environment have been published [19,20]. Most knowledge pertaining to the presence of *M. ulcerans* in the environment stems from quantitative polymerase chain reaction (qPCR) analyses [21,22]. *M. ulcerans* DNA has been detected in many types of environmental samples, including aquatic invertebrates, fish, aquatic plant biofilms, plant rhizospheres, water filtrates, and soil [23]. However, for unknown reasons, reported PCR positivity from environmental studies in settings that are endemic for Buruli ulcer in Africa differ substantially. Nonetheless, findings indicate that *M. ulcerans* can persist in aquatic environments such as wetlands or side streams of rivers [24] and, indeed, aquatic reservoirs appear to play an important role in transmission [25–27]. In the south-eastern part of Australia, possums have been identified as animal reservoir [28,29]. In Africa, grasscutters (*Thryonomys swinderianus*) have been found to be susceptible to *M. ulcerans* infection [30]. It is suggested that inoculation of *M. ulcerans* underneath the skin either through mechanical injury or through insect bites as mechanical vectors may be relevant for transmission of *M. ulcerans* from environmental reservoirs [22,23]. Understanding the route of transmission is critical to protect endemic communities.

Prior studies investigated several risk factors that might facilitate the transmission of Buruli ulcer. For example, quantitative studies revealed that contact with unprotected fresh water bodies is associated with a high number of Buruli ulcer infections [16,22,23,31]. Stagnant and unprotected fresh water sources seem to present the highest risk [26]. Swimming in stagnant water has been perceived as source of Buruli ulcer [15]. In a case-control study in Benin, it was shown that regular use of water from new wells protected against *M. ulcerans* infection [32]. Interestingly human-induced changes of the environment, such as the construction of dams, large-scale irrigation schemes, and mining might increase the incidence of Buruli ulcer [17]. A deeper understanding of the environment-to-human transmission may lead to the identification of preventable risk factors and to recommendations, such as the use of adequate waterproof protective equipment during agricultural activities and water, sanitation, and hygiene (WASH) interventions. Community-based approaches are required to reveal the broad spectrum of water-related activities.

We aimed to better understand the water contact patterns of Buruli ulcer cases by matching geo-referenced and qualitative data. The following objectives guided our study. First, to map water resources contacted by patients with Buruli ulcer. Second, to assess the diversity and characterize surface water bodies frequented by Buruli ulcer patients. Third, to better understand how Buruli ulcer cases might be exposed to different surface water bodies identified.

## Methods

### Ethics statement

The study received ethical approval from the National Committee for Ethics in Life Science and Health of Côte d'Ivoire (no. 025-22/MSHPCMU/CNESVS-km). Participants were informed about the objectives and methods and what was expected of them by clarifying their roles in the study. Participation was voluntary, and hence, participants could withdraw from the study anytime without affecting their access to services offered by the health facility. All participants or their parents/care taker in case of children signed a written informed consent as part of the larger wound management project. In addition, oral consent was obtained from all participants for the current sub-study. All identified Buruli ulcer cases received standard chemotherapy (i.e., rifampicin at 10 mg/kg body weight and clarithromycin at 7.5 mg/kg body weight for eight weeks) and wound management for Buruli ulcer, adhering to national guidelines.

### Study set-up and participants

This study was part of a broader research project, aiming to reduce the burden of chronic, complicated wounds in the south-central part of Côte d'Ivoire. The "wound management project" applies a community-based approach to identify wounds early and to treat them broadly. Since the project was launched officially in the study area in 2019, the project has engaged community health workers, who have established close contact with the local population [33,34]. For our study, cases that were tested positive for Buruli ulcer by IS2404 PCR between May and September 2019 were eligible to participate. After laboratory confirmation by the national reference center in Abidjan (Institut Pasteur de Côte d'Ivoire at Cocody CHU), identified Buruli ulcer cases from the two study villages Ahondo and Sahoua were invited to participate. Ahondo and Sahoua make up the Ahondo Health area, which has been the focal area of the framing research project. Since 2019, standardized case report forms are utilized for tracking the wounds and active case detection was performed in both villages through household surveys in 2019 (May-July) and early 2022 [33]. Notably, both villages along with another 11 villages, form the Taabo health and demographic surveillance system (HDSS) [35].

The climate in the Taabo HDSS is tropical with a long rainy season from April to July and a shorter rainy season in October and November. Main sources of income are farming and fishing, including the cultivation of cacao, coffee, roots, tubers, banana, and rice. In the late 1970s, a large dam was constructed across the Bandama River in Taabo for hydroelectric power production. The impounded lake extends over nearly 70 km$^2$. Less than 20 km upstream of the dam, the two study villages, Ahondo and Sahoua, are located in close proximity to the river (Fig 1). In both villages few pumps, wells, and private taps are installed, yet comprehensive and reliable water infrastructures remain an issue. In 2019, 3,021 and 1,167 residents lived in Ahondo and Sahoua, respectively, according to the Taabo HDSS database. In Ahondo, there is a community health center and in Sahoua a dispensary, offering essential health care services to residents. In both localities, community health workers visit patients at home to take care of the wounds. Additionally, nurse assistants provide wound management at the health care facilities. Both health care facilities are supported by the referral Taabo General Hospital [36].

### Data collection

In Ahondo and Sahoua, community health workers, together with a research assistant, acted as gate-keepers and facilitated the introduction to the targeted laboratory-reconfirmed former Buruli ulcer cases. For our geospatial qualitative inquiry, data were collected in two steps. In a

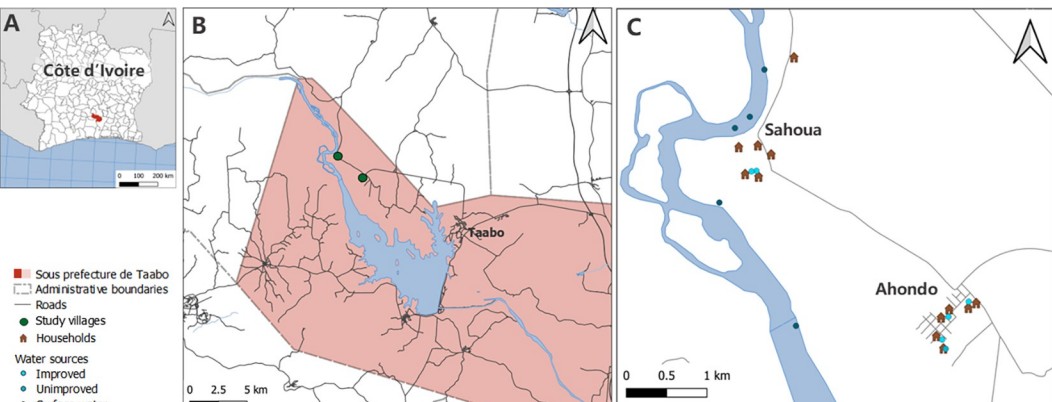

**Fig 1. Map indicating the study area.** (A) Côte d'Ivoire; (B) "sous-prefecture de Taabo"; and (C) the location of households and type of water sources of Buruli ulcer cases in the two study villages Ahondo and Sahoua. Base-layers retrieved from https://www.diva-gis.org/gdata.

first round of data collection, the selected Buruli ulcer cases were visited at home in December 2021 and January 2022 and invited to indicate the location of water sources they contacted, which were subsequently visited by the researchers. Observation of the visits were documented in a field journal and photographs. Global positioning system (GPS) data from the households and water sources were captured with MAPS.ME (MAPS.ME offline application based on OpenStreetMap data).

Based on this initial geospatial inquiry, the same cases were re-visited in March 2022 for in-depth interviews about water-related activities and perceived transmission pathways. To complement the perspective of the cases, interviews with health care workers were conducted about perceived risk factors and the distribution of Buruli ulcer cases. After training, piloting, and validation of the questionnaire, a research assistant conducted all interviews. Interviews were administered in French and, when need be, complemented with local languages (i.e., Baoulé and Malinké). Interviews with Buruli ulcer patients were voice recorded and notes were taken in a structured tabular format. Verbatim transcripts were obtained from interviews with health care workers.

## Data management and analysis

Geo-referenced household data and water sources were mapped and visualized in QGIS (QGIS, 2022; QGIS Geographic Information System, QGIS Association; http://www.qgis.org). Descriptive statistics of study participants (i.e., Buruli ulcer patients and health care workers) were calculated using STATA (Stata Corp, 2019; College Station, Texas, United States of America). Notes and transcripts from the interviews were managed and analyzed in Nvivo, a qualitative data analysis software (Nvivo 12 Pro. QSR International, 2021; https://www.qsrinternational.com/nvivo-qualitative-data-analysis-software/home). The coding system for thematic analysis was closely linked to the different topics addressed in the questionnaire [37]. Emerging sub-codes were added during the analysis. A separate code was used to extract illustrative quotations. Water sources were categorized using the terminology of the WHO/UNICEF Joint Monitoring Programme (JMP) [38].

## Results

### Demographic characteristics of participants

As summarized in Table 1, six Buruli ulcer cases from each of the two villages (i.e., Ahondo and Sahoua) participated in the study, making up a total of 12 participants (seven males and

**Table 1. Demographic characteristics of Buruli ulcer cases ($n$ = 12) included in this study from south-central Côte d'Ivoire in late 2021/early 2022.**

|  | Ahondo | Sahoua | Total |
|---|---|---|---|
| **Number of interviewed cases** |  |  |  |
| Female | 3 (50%) | 2 (33.3%) | 5 (41.6%) |
| Male | 3 (50%) | 4 (66.7%) | 7 (58.3%) |
| **Total** | **6 (100%)** | **6 (100%)** | **12** |
| **Average age (and range) in years** |  |  |  |
| Female | 21 (7–43) | 28 (11–45) | 24 (7–45) |
| Male | 11 (6–19) | 38 (15–60) | 26 (6–60) |
| **Total** | **16 (6–43)** | **26 (11–60)** | **25 (6–60)** |
| **Average duration (and range) living in the community in years** |  |  |  |
| Female | 21 (6–43) | 25.5 (6–45) | 23 (6–45) |
| Male | 10 (5–19) | 18 (7–27) | 14.5 (5–27) |
| **Total** | **15.5 (5–43)** | **20.5 (6–45)** | **18 (5–45)** |

**Table 2. Demographic characteristics of health care workers (*n* = 4) included in this study from south-central Côte d'Ivoire in late 2021/early 2022.**

| | Sex | Age (in years) | Years living in the community | Years at practice |
|---|---|---|---|---|
| **Ahondo** | | | | |
| Nurse assistant | M | 36 | 5 | 5 |
| Community health worker | F | 39 | 39 | 14 |
| **Sahoua** | | | | |
| Nurse assistant | M | 35 | 2 | 2 |
| Community health worker | F | 42 | 42 | 13 |
| **Total (average)** | | **38** | **22** | **8.5** |

five females). The average age of participants was 25 years, consisting of five children aged below 15 years (41.7%), and seven adults (58.3%). Most Buruli ulcer cases were living in the village since they were born. All except one Buruli ulcer lesion were located on the extremities (10 on lower legs, one on forearm). The remaining wound was located on the chest of a patient. All lesions had ulcerated, requiring treatment at the Buruli ulcer clinics in Taabo. After initial wound management in Taabo, patients returned to their villages and were followed-up by community health workers. All wounds were healed at the time of the follow-up interviews.

Four health care workers (two from each village), who were part of the framing wound management project, were interviewed (Table 2). Notably, both community health workers were born in the respective village and have served the communities since more than 10 years.

## Geographic location and spatial distribution of Buruli ulcer cases' households

Based on household mapping, no hot spots of cases within the communities were observed (Fig 1). However, most cases reported that close relatives were also infected, indicating micro-level clustering. Whilst talking about the development of the wounds, several interviewed cases mentioned that these kind of "bad wounds" ("*Kani tèh*"), as they describe it in the local language (Baoulé swamlin), repeatedly also affected others in their community. Furthermore, almost half of the participants emphasized the irremediable or even fatal consequences of an infection, which is illustrated in the following quote from a health care worker:

> *"In the [patient's] yard there, almost everyone had Buruli ulcer. It was this kind of wound that killed his mama. Those wounds killed his little brother. The wounds killed his big sister's child. [Name] had Buruli ulcer, there's [name] who had Buruli ulcer, there's the son who had Buruli ulcer. Huh! She even had Buruli ulcer. Almost everyone in their yard."* (CHW_2)

## Water sources contacted

In both villages, similar water sources were reportedly contacted, spanning from unprotected surface water bodies to improved water sources (Table 3). Snapshots of the water sources and related activities are provided in Fig 2. In Ahondo, only few participants had their private tap in close proximity to their house, which was connected to a drilled well. In both villages, public taps connected to a drilled well equipped with motor pumps and storage tanks were available (Fig 2C). Notably, fetching water at these improved water sources is subject to a small fee. In Sahoua, also mechanical boreholes (i.e., hand pumps) were frequented. However, several participants stated that they do not fully rely upon these wells and pumps, as there are times during the year they lack financial means or taps and pumps might not be working. During the

**Table 3. Overview of water resources frequented by Buruli ulcer cases interviewed in the study area in south-central Côte d'Ivoire in late 2021/early 2022.**

| Water resource | Key characteristics | Related activities |
|---|---|---|
| **Improved** | | |
| Water tower and taps | Water tower connected to several taps in the village | Drinking water; washing and bathing depending on availability (or nearby natural sources) |
| Drilled well and taps | Motorized pump connected to nearby tank and taps, often charged (or mechanic pump) | Drinking water; washing and bathing depending on availability (or nearby natural sources) |
| Rain water | Rain water collected and stored at home | Drinking water; washing and bathing depending on availability (or nearby natural sources) |
| **Unimproved** | | |
| Dug wells (unprotected) | Whole dug in the ground | Drinking water; washing and bathing depending on availability (or nearby natural sources) |
| **Surface water** | | |
| River | Nearby riverbanks of the Bandama River | Laundry, dish washing, bathing, playing, fetching water, watering agricultural fields, mud brick making |
| Low-lands and ponds ("*bas fonds*" and "*marigots*") | Natural and seasonal low-lands and ponds, filled with rain water | Laundry, dish washing, bathing, playing, fetching water, watering agricultural fields |

Water resources were classified according to the WHO/UNICEF Joint Monitoring Programme (JMP) drinking water ladder

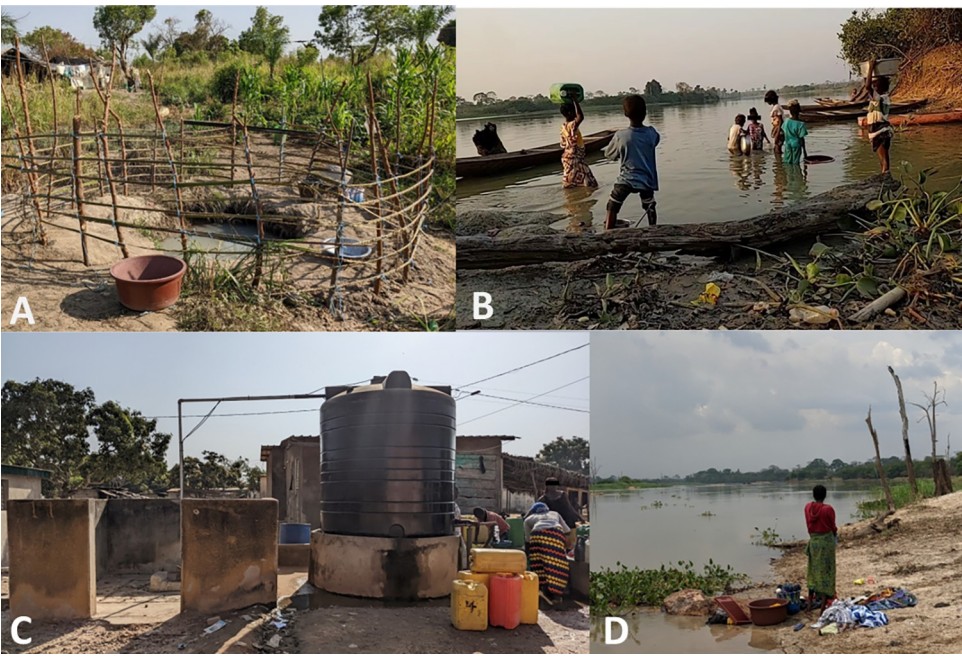

**Fig 2. Type of water resources and typical human-water contacts in the study area in south-central Côte d'Ivoire in late 2021/early 2022.** (A) dug well; (B) children fetching water at the riverbank; (C) drilled well with pump, reservoir, and tap; and (D) woman washing clothes at the river.

rainy season, multiple participants reported to collect rain water from their roofs in basins or barrels for domestic use.

As unimproved water sources, few participants dug their own wells in close proximity to their house (Fig 2A). Water holes visited were usually not protected. Natural occurring surface water bodies were also reported as important sources by cases and health care workers. For all participants, the Bandama River was a key water source throughout the year, but especially during the dry season (Fig 2B and 2D). Furthermore, nearly all participants use seasonal low-lands ("*bas fonds*") and ponds ("*marigots*"), which are-when filled with water-closer to their houses than the river. Health care workers suspected contact with these natural water sources (river, low-lands, and ponds) to increase the risk of infection with *M. ulcerans*.

## Water-related activities and exposure to surface water bodies

As summarized in Table 3, a broad range of activities were reported to be undertaken by or with surface water, including water from low-lands, ponds, and the river. Participants described that these natural water bodies are used for a multitude of housekeeping activities, such as laundry, dishwashing, and bathing. Low-lands and natural ponds are also used for watering agricultural fields, as participants explained. Some participants had their fields on the other side of the river, and hence, cross the river in a wooden canoe ("*pirogue*") to get there. When crossing the river, participants like to drink the water, preferable from the middle of the stream. Once at the riverbank, many participants also like to swim. Children like to play in the water at the riverbanks and in the seasonal ponds. When visiting the riverbank, women, who are traditionally in charge of washing, were observed doing their laundry using buckets, basins, and washboards (Fig 2C). Children and women were also there to fetch water (Fig 2B). According to some participants, fetching water at the river in large quantities, transporting it with a rented vehicle to their homes and conserve it in either plastic, iron, or cement barrels is frequently done. One participant also mentioned the use of water from the river to produce mud bricks, which are used for their own purpose or to be sold.

The frequency and duration of water exposure varied between participants. For example, a farmer from Sahoua went to his rice fields behind the river daily, where he is working in the swamp for several hours a day. Others only went to their field during the harvesting season to support the families' agricultural production. Some children have to cross a low-land, which is seasonally filled with water, to go to school. Many women reported that they frequent the river daily for washing and fetching water, which they carry home on their heads or by bicycle. The mother of a small boy said that her son is barely in direct contact with water as the river is too far and too dangerous for children. Yet, he and his siblings contracted Buruli ulcer. In contrast, two cases reported that since they contracted the disease, they avoid stepping into the river.

## Perceived elevated risk factors

Beyond specific patterns of water use, several environmental, structural, and individual factors emerged that are perceived as elevated risk factors.

**Environmental factors.**   During the interviews, participants differentiated flowing and stagnant surface water bodies. Stagnant water is perceived as dirtier and containing more microbes. In turn, flowing water is considered cleaner, as one patient with Buruli ulcer said:

*"When [the water] flows, it is clean, there is no rubbish on it." (S12_01)*

Despite these manifestations, the interviewees prefer water sources near their homes if they are providing sufficient water yield. Hence, (seasonal) availability and proximity seem to be

more important for selecting a water source, than water quality, especially when water is needed in large quantities for example for watering agricultural plots.

Most participants noted a relationship of surface water and presence of insects. Different insects were mentioned, including water bugs, tse-tse flies, and "*les moucherons*" (midges). However, there was no clear consensus about the exact water sources, season, and type of insects. Some participants suspect that insects can transmit Buruli ulcer, as illustrated in the following quote:

*"Particularly when it rains, the insect is on the water. At the same time, it sticks on the plants. When someone is walking by, it attaches to the clothes or the skin [. . .]. It stings as self-defence and at the same time ejects eggs, these eggs, and then it just begins." (S10_01)*

Regarding riverbanks, participants reported both sand and mud banks. Sand banks were preferred as people feel safer on the solid ground. Mud banks were perceived more dangerous, as they feared to sink in. Furthermore, when speaking about the muddy riverbanks, participants referred to the local expression "*poto-poto*", indicating the dirty, obscure ground around natural water bodies:

*"The water there, there is "poto-poto" on the ground [. . .], black mud [. . .] maybe there is also spiky garbage on the ground." (S13_01)*

**Structural and social factors.** Different factors on the communal or societal level influence the nature of water exposure of residents. For example, depending on the availability, functionality, and reliability of water infrastructure (taps and pumps), people may depend more on improved than surface water sources. Some health care workers indicated that they believe that, in particular, the availability and reliability of running water in the community is a key factor influencing the incidence of Buruli ulcer. Additionally, water contact patterns were governed by gender division of labor. When asked about their patterns of contacting natural water sources, males mentioned primarily agricultural chores. Females, on the other hand, reported mostly specific household activities and specific care work for their children.

**Individual factors.** Participants perceived individual factors associated with an elevated risk for contracting Buruli ulcer. For instance, most of the participants enter barefoot or with plastic sandals into the water. Based on participants' statement, protective equipment (boots, gloves, etc.) is therefore rarely used during agricultural activities. Only one participant who is working in a nearby industrial banana plantation site is wearing boots, as requested by the company.

Moreover, financial means determine where to seek water, especially water from drilled holes as there is a fee. The choice of water is also influenced by personal preferences, as residents seem to prefer the natural water due its specific taste. Some participants said that they particularly like to drink rain water and the water from the river. Others frequent the river because they are used to it, as explained by a health care worker:

*"I can say that it is a question of habit. The parents are used to drink the water of the river. We can only tell them not to drink the water from the Bandama or the ponds. That's it! Otherwise, we can say in Sahoua there are many pumps, but as it is a habit for people to drink other waters [. . .] we cannot change it." (CHW_1)*

When fetching water at surface water bodies, exposure depends also on the type of container that the people employ. Based on our observation at the river as well as at the

households, typical containers for fetching water are buckets ("*des seaux*"), basins ("*des cuvettes*"), or cans ("*des bidons*"). Buckets are usually filled in one go. Basins are put on the head (half filled) and then filled up by pouring water with a cup in the basin. Cans are plunged into the water. Participants described that the deeper the water, the faster the can is filled and the less far and deep one must stand in the water.

## Discussion

Our qualitative investigation in two Buruli ulcer-endemic villages in the south-central part of Côte d'Ivoire shows a wide range of water-related domestic activities that might expose residents to the risk of contracting certain water-borne diseases and specifically Buruli ulcer. Our research highlights a considerable variety of water-related activities, exposing local residents to flowing but also stagnant water sources. Be it for personal household chores, agricultural activities, or other income-generating tasks, natural water sources, including the Bandama River, seasonal low-lands and ponds, were frequented on a daily basis as the participants reported. At times, these are the only available water sources in the study villages. Overall, water contact patterns observed and reported by the interviewees are in line with case studies from similar settings in West Africa [25,32]. To compare and contrast perceived risk factors more comprehensively, it would be interesting to investigate water contact patterns longitudinally in multiple sites considering households with and without Buruli ulcer cases.

We did not find specific micro-clustering of Buruli ulcer cases in the study settlements; yet, certain characteristics emerged that are offered for consideration. First, five of the 12 Buruli ulcer cases (41.7%) were children (<15 years). Although the small size of study population does not allow generalization, it should be noted that Buruli ulcer predominately affects children [3]. Second, when comparing the number of cases included in our study to the total population, relatively more Buruli ulcer cases came from Sahoua (6 out of 1,167; 0.51%) compared to Ahondo (6 out of 3,021; 0.20%). In line with findings from household wound surveys from the framing research project [33] our qualitative inquiry offers indication that Buruli ulcer remains endemic in the Taabo HDSS [39]. Third, many of the Buruli ulcer patients interviewed mentioned that relatives suffered from similar "bad wounds". This underlines the call to scrutinize family histories [40]. Most notably, Buruli ulcer cases were also associated to fatal consequences. Hence, there is a pressing need to investigate this issue. As the two study villages are part of the Taabo HDSS, key demographic and health variables are monitored longitudinally, such as pregnancy, birth, in- and out-migration, and death. As regards the latter, causes of death are investigated, using verbal autopsies [41]. Fourth, despite the small sample size, but in line with recent findings elsewhere [39], we hypothesize that Buruli ulcer cases are considerably underreported, as compared to the incidence reported by WHO [2].

Consistent with previous research, our study confirms that the affected population perceives that aquatic environments play a central role in the transmission of *M. ulcerans*. Many participants suspected the river and stagnant water sources to cause the disease and were particularly afraid of being bitten by insects. However, all interviewees depend on the water for different kinds of domestic, recreational, and subsistence use. This is in line with findings from similar settings in other West African countries, where Buruli ulcer-endemic areas are located in close proximity to major river valleys (or dams). For example, while investigating the epidemiology of Buruli ulcer in Cameroon, the Mape Basin and Nyong River Valley emerged as key transmission settings [5,42]. In the south-western part of Nigeria, Buruli ulcer studies were conducted with cases in the Cross River State [43]. In Ghana, several studies were carried out in the Densu River Basin [44] and the Offin River [27,45]. In Benin, multiple studies were done in the Ouémé and Plateau districts, including seminal investigations of the

impacts of water sources on Buruli ulcer [25,32]. A recent study suggests that improved water infrastructure is associated with lower number of Buruli ulcer cases [32]. In line with these observations, our qualitative study underlines the diversity of human-water contact patterns, as determined by environmental, structural, social, and individual factors [25]. Indeed, subsistence farming and likely also mining play a key role in the water exposure as well as risk activities for wounds, as shown by research from Cameroon [46]. Our data indicate gender-specific exposure mechanisms likely linked to the social division of labor. Hence, it would be interesting to investigate these gender-specific water contact patterns in greater depth, as different population groups (e.g., women and children) are exposed to different water sources. New research is needed to unravel the complexity of water-related exposure mechanisms and to better understand the social-ecological contexts [40].

Integrated disease management is proposed to reduce the burden of poverty-related diseases in a comprehensive manner, including Buruli ulcer [7,33]. This warrants close collaboration with affected communities, including training for wound management for dressing and recognizing the need to refer a patient to the hospital. Despite the larger project-related progress in the Taabo HDSS, the health care workers interviewed in our study emphasized that additional sensitization is still needed to further increase awareness, improve current knowledge, and reduce the stigmatization of Buruli ulcer among the study population. Particularly the latter could be helpful to address the persisting beliefs, attributing Buruli ulcer to metaphysical factors, witchcraft, and spells [12]. Including the community perspective to further investigate the complex interplay of environmental, cultural, and behavioral risk factors is therefore inevitable. Especially culturally appropriate and effective interventions, which can be easily adopted by affected communities, should be designed and implemented. Indeed, community engagement is key to actively prevent Buruli ulcer and allowing them to contribute the early identification of cases. Besides community-based interventions, the community perspective is important to inform large-scale environmental sampling. Furthermore, community perspective might guide the identification and monitoring of water sources over time [15]. Stakeholders from different disciplines are needed to reveal the transmission of the mysterious disease and reduce the burden of the "bad wounds". One Health approaches, as fostered by the wound management project, shall continue to address Buruli ulcer in a holistic manner, including poverty, malnutrition, and inequities, which are at the roots of neglected tropical diseases.

Our study has several strengths and limitations that are offered for discussion. First, it is important to note that our study was carried out in the Taabo HDSS, and hence, there is a history of conducting research projects since about 15 years on top of longitudinal monitoring of key demographic events and investigating causes of death [35,41,47–52]. A comprehensive wound management project, launched in 2019, provided the frame for the current study. Prior research and interventions certainly increased awareness among residents and might have altered health seeking behavior, compared to less researched areas. Hence, our findings might not be representative for other rural areas of Côte d'Ivoire and elsewhere in West and Central Africa. Second, our study focused on Buruli ulcer cases from two endemic villages. The sample size with only 12 cases was limited and no control villages were included. Of note, data saturation was attained by the 12 interviews and cases' perspectives were complemented by interviews with health care workers. It would have been interesting to triangulate our findings with secondary data such as verbal autopsies of the Taabo HDSS. Third, our study was implemented during the dry season (December 2021 to March 2022). Findings must therefore be interpreted with caution, as water exposure patterns might look considerably different during the rainy season. Observations over at least one year are warranted, placing particular emphasis on human-water contact patterns.

## Conclusion

Buruli ulcer is a neglected tropical skin disease that remains endemic in the Taabo HDSS in south-central Côte d'Ivoire. Our qualitative investigation in two villages underlines the frequent use of unprotected surface water, also in places where drilled wells with pumps are installed. A multitude of water-related activities occur, leading to sometimes very intense water contact. Hence, people are at risk of contracting Buruli ulcer, as confirmed here. Besides these environmental aspects, the exposure is interlinked with cultural, social, and behavioral factors. Our findings therefore underline the importance of integrating local knowledge and contexts in future studies to unravelling the transmission of *M. ulcerans*. A deeper understanding of human-water contact patterns of affected communities is important to inform large-scale environmental sampling, the design of interventions, or disease-specific health promotion campaigns. While official Buruli ulcer case numbers might be underreported, this study specifically raises the voice of affected populations, calling for increased awareness and community engagement to combat this neglected tropical skin disease and reduce its health, social, and economic burden.

## Acknowledgments

This study has been realized thanks to the close collaboration between several institutions and projects. We are grateful for the logistical and infrastructural support of the Centre Suisse de Recherches Scientifiques en Côte d'Ivoire, Afrique One-ASPIRE (DEL-15-008), and the wound management project that is financially supported by the Else Kröner-Fesenius-Stiftung (project no. 2017_HA19). We thank Prof. Thomas Junghanss and Dr. Marija Stojkovic for their sedulous support and guidance to realize the study. We are indebted for the collaboration with the national control program against Buruli ulcer. We thank everyone from the wound management project team in Taabo and the health care workers from Taabo, Ahondo, and Sahoua. We are grateful for our research assistants for the data collection and transcription. Special thanks go to all study participants for sharing their time and experiences so abundantly with us.

## Author Contributions

**Conceptualization:** Andrea Leuenberger, Raymond T. A. S. N'krumah, Didier Y. Koffi, Bassirou Bonfoh, Jürg Utzinger, Gerd Pluschke.

**Data curation:** Andrea Leuenberger.

**Investigation:** Andrea Leuenberger, Bognan V. Koné.

**Methodology:** Andrea Leuenberger, Bognan V. Koné, Raymond T. A. S. N'krumah, Didier Y. Koffi.

**Project administration:** Bognan V. Koné.

**Supervision:** Bassirou Bonfoh, Jürg Utzinger, Gerd Pluschke.

**Visualization:** Andrea Leuenberger.

**Writing – original draft:** Andrea Leuenberger, Bognan V. Koné, Jürg Utzinger, Gerd Pluschke.

**Writing – review & editing:** Andrea Leuenberger, Bognan V. Koné, Raymond T. A. S. N'krumah, Didier Y. Koffi, Bassirou Bonfoh, Jürg Utzinger, Gerd Pluschke.

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
