## [Decision Letter · Decision Letter 0]

13 Sep 2022

Dear Dr. Leuenberger,

Thank you very much for submitting your manuscript "Perceived water-related risk factors of Buruli ulcer in south-central Côte d’Ivoire" for consideration at PLOS Neglected Tropical Diseases. As with all papers reviewed by the journal, your manuscript was reviewed by members of the editorial board and by several independent reviewers. In light of the reviews (below this email), we would like to invite the resubmission of a significantly-revised version that takes into account the reviewers' comments. 

Dear Dr. Leuenberger,

Your manuscript has been reviewed by three experts in the field. As you can see there is a wide range of issues to be addressed and suggestions of where the manuscript should be revised. Please address all the criticisms and provide more details about the subjects lesions and also discuss the obstacles to more effective implementation of health education measures.

We cannot make any decision about publication until we have seen the revised manuscript and your response to the reviewers' comments. Your revised manuscript is also likely to be sent to reviewers for further evaluation.

Sincerely,

Paul J. Converse

Academic Editor

Ana LTO Nascimento

Section Editor

Dear Dr. Leuenberger,

Your manuscript has been reviewed by three experts in the field. As you can see there is a wide range of issues to be addressed and suggestions of where the manuscript should be revised. Please address all the criticisms and provide more details about the subjects lesions and also discuss the obstacles to more effective implementation of health education measures.

Reviewer's Responses to Questions

**Key Review Criteria Required for Acceptance?**

**Methods**

-Are the objectives of the study clearly articulated with a clear testable hypothesis stated?

-Is the study design appropriate to address the stated objectives?

-Is the population clearly described and appropriate for the hypothesis being tested?

-Is the sample size sufficient to ensure adequate power to address the hypothesis being tested?

-Were correct statistical analysis used to support conclusions?

-Are there concerns about ethical or regulatory requirements being met?

Reviewer #1: This is a qualitative study aiming to understand perceptions about Buruli ulcer transmission.

It did not aim to identify any association between behaviour and disease, which would need a much larger sample and a control group.

Reviewer #2: The report of this study is part of a major study being undertaken in the study area. This is well described in the manuscript and the objectives for this qualitative study and how it relates to the larger study clearly outlined. The Prefecture and communities selected for the study are reported nationally as BU endemic areas making them appropriate for such a study. Prevalence of confirmed Buruli ulcer in communities is generally low, therefore, although the sample size is small, it is appropriate for the study being undertake, being qualitative. The information being solicited for could only be give by this sample size and therefore only percentages of demography was performed as a statistic. Lesion types were however not described in the sample as well as exact parts of the body where the lesions were located (extremities). This could be shown in a table.

Regulatory and ethical requirements were met as the study was within the larger project for which ethical clearance has been granted. The study however failed to mention treatment of the patients and the confirmation laboratories used.

Reviewer #3: Here is a qualitative study that aims to investigate the perceived water-related risk factors of Buruli ulcer in south-central Côte d'Ivoire

Even if it is a qualitative study, the number of subjects included seems to us to be very unrepresentative of the south-central Ivory Coast. (12 subjects included in 2 villages) How did you choose these two villages? How many villages does Taabo district have?

What is the WASH infrastructure coverage?

Is there a reason why the authors did not organize focus group discussions in order to compensate for the relatively small number of subjects included?

 I therefore suggest reviewing the title and clarifying the perceived water-related risk factors of Buruli ulcer in the localities investigated.

**Results**

-Does the analysis presented match the analysis plan?

-Are the results clearly and completely presented?

-Are the figures (Tables, Images) of sufficient quality for clarity?

Reviewer #1: The results are clearly stated.

Reviewer #2: The objective of the study was to elicit for perceived risks of BU infection in the study communities and it targeted only reconfirmed patients. This could be described as preliminary to further investigation into the transmission of the Mycobacterium ulcerans and its reservoirs. The selection and interviews of the patients produced the required information. In the analysis, a better description of the exact positions and sizes of the lesions or nodules could have supported the exact risks to infection (socioeconomic activities, recreational activities, etc). Photos of the lesions should be presented to support selection of cases (if the ethical clearance allows).

In the section from line 216 (Spatial Distribution of BU cases), the low number of cases in this study provides a challenge to determining spatial distribution of cases. The title of this section should be edited to reflect the discussion of household cases as in the text.

Healthcare workers were interviewed but little is reported of the results of those interviews and what it meant to the study or the information gathered from them.

In the discussion (lines 368 – 371), the authors estimated the prevalence as a percentage for the communities based on the sample size. The description there is problematic as they failed to use the actual cases over a specific period of time for these estimates. Moreover, the authors could have easily verified the claim that Buruli ulcer has been fatal to some family members of the participants. The HDSS system in Taabo could provide this information.

Reviewer #3: The results of the study are certainly interesting but do not really bring new data.

In our opinion, the results must be better studied in order to bring out the deep mechanisms underlying the choices of the populations and to see with them the levers on which we can base ourselves to change the situation within the framework of a "problem solving" approach.

Can the authors give us an overview of the number of BU cases in the districts. and in the villages included over time ?.

**Conclusions**

-Are the conclusions supported by the data presented?

-Are the limitations of analysis clearly described?

-Do the authors discuss how these data can be helpful to advance our understanding of the topic under study?

-Is public health relevance addressed?

Reviewer #1: The main message is stated as: "this study specifically raises the voice of affected populations, calling for increased awareness and improved knowledge to combat this neglected tropical skin disease and reduce its health, social, and economic consequences." I disagree with this - I think that the study shows that knowledge and awareness about BU are quite well developed in these villages, where various projects have been operating for at least 20 years (I have been there myself several times!). The key issue is that people are not motivated to put this knowledge into practice. This is the fundamental obstacle faced by all programs of health education or health promotion. Reorienting the Discussion to this problem would make the paper more insightful.

Reviewer #2: The conclusions presented in the manuscript reflect the information sought by the study and the objectives. Results from this study confirms the need to map environmental contacts communities when undertaking ecological and transmission studies of Mycobacterium ulcerans. The authors also used information gathered in the interviews to suggest the need for tailormade designs for interventions in Buruli ulcer endemic communities.

Reviewer #3: No specific comments

**Editorial and Data Presentation Modifications?**

Reviewer #1: Revision of the Discussion would be helpful.

Reviewer #2: To improve the paper, the authors:

1. Improve on their case selection criteria by describing the lesions and adding photos.

2. Edit title on line 216 to reflect content of paragraphs

3. Adjust estimates of prevalence with actual case numbers and not only samples

4. Add information gathered from health workers

5. Verify fatality of Buruli ulcer in the DHSS.

Reviewer #3: (No Response)

**Summary and General Comments**

Reviewer #1: Revision of the Discussion would be helpful.

Reviewer #2: This paper reports on a very pressing public health problem and has added valuable information on the importance of environmental mapping in addressing the transmission issues of Buruli ulcer. Buruli ulcer remains a disease of public health importance and the selection of the study communities is appropriate to the study. The authors also clarified that this study was part of a larger study. A main weakness of this paper is the number of cases that were interviewed, but being largely qualitative, information gathered from the patients is valuable and its use appropriate to the objectives of the study.

Reviewer #3: In the current state of the manuscript, it seems to us that the manuscript requires a major revision. The methodology and results section deserve to be better explained and more detailed.

Failing this, the authors must review the title and the objectives of the study in order to report a preliminary study to be deepened by subsequent investigations.

PLOS authors have the option to publish the peer review history of their article (what does this mean?). If published, this will include your full peer review and any attached files.

Reviewer #1: Yes: Paul Saunderson

Reviewer #2: No

Reviewer #3: No
---

## [Editor Report · Decision Letter 1]

1 Nov 2022

Dear Dr. Leuenberger,

We are pleased to inform you that your manuscript 'Perceived water-related risk factors of Buruli ulcer in two villages of south-central Côte d’Ivoire' has been provisionally accepted for publication in PLOS Neglected Tropical Diseases.

Best regards,

Paul J. Converse

Academic Editor

Ana LTO Nascimento

Section Editor

Thank you for your careful and thorough responses to the concerns of the reviewers.

---

## [Editor Report · Acceptance letter]

30 Nov 2022

Dear Ms Leuenberger,

We are delighted to inform you that your manuscript, "Perceived water-related risk factors of Buruli ulcer in two villages of south-central Côte d’Ivoire," has been formally accepted for publication in PLOS Neglected Tropical Diseases.

Best regards,

Shaden Kamhawi

co-Editor-in-Chief

Paul Brindley

co-Editor-in-Chief
